# Land Development Planning: New Principles and New Representations in the General Urban Plan of Parma

**Chiara Vernizzi * and Chiara Finizza**

Department of Engineering and Architecture, University of Parma, 43124 Parma, Italy; chiara.finizza@unipr.it
* Correspondence: chiara.vernizzi@unipr.it; Tel.: +39-3475752442

**Abstract:** Since the entry of Emilia-Romagna Regional Law No. 24/2017 *Disciplina regionale sulla tutela e l'uso del territorio*, the processes guiding the development of the territory have profoundly changed, in principle, in the structuring through processes of participation and how contents are expressed and represented. Concepts such as the containment of soil consumption as a non-renewable resource; regeneration of urbanized territories; protection and enhancement of land, including agricultural land; protection and enhancement of historical and cultural elements; and promotion of greater levels of knowledge of the territory and its existing built heritage are the main principles of the law that give foundational importance to the entire process of forming the knowledge framework and strategic lines that structure General Urban Plans, which are designed to express planning contents through an ideogrammatic type of cartography. The analysis of the law and its contents also becomes an opportunity to reflect on the tools through which the governance of the territory is implemented, from GIS to web GIS up to the changes in the graphic language through which the planning principles are expressed, with specific reference to the planning tools of Parma, on which the new General Urban Plan (PUG) is being defined.

**Keywords:** urban planning; cartographic representation; ideograms; web GIS; interactivity maps; participatory process





## 1. Introduction

This contribution intends to investigate the role of representation in urban planning, with particular reference to the relationship between the way in which the graphic symbols used are related to the different ways of planning city growth, through a temporal excursus from the first plan elaborated for the city of Parma in 1887 to the recently assumed PUG by the Municipality of Parma. Thus, the representation is implemented in a different way than in the past, as it has to adapt to constantly and very rapidly changing dynamics [1] (p. 567). The study of the semantic value of signs and symbols used in urban planning is still a relatively unaddressed field, while issues concerning the technological tools for supporting and sharing information (e.g., interactive maps) are a topic that has undergone great evolution in the last decade. In the definition of the most suitable and effective information tools for communicating technical content, the role and meaning of the graphic signs used, which today have an ideogrammatic role (as seen in the case study described above), are expressed in relation to the type of vision that the new town planning instruments of the Emilia-Romagna Region are required to construct in order to flexibly articulate and describe the future growth strategies of the cities in the territory.

The topic, therefore, is not so much about solving planning problems but is aimed at providing and illustrating how representation has changed its expressive language. Indeed, there is no more suitable tool for understanding and describing the physical component of the landscape than drawing. On the other hand, the urban landscape today is so complex that it needs to be approached in a multidisciplinary way and structured as an "open system of knowledge" [2] (p. 6).

Consequently, it is legitimate to reflect on the potential of these new approaches to landscape representation and communication. The need to manage and represent the complexity of urban space, as well as the need to process and communicate the increasing amount of information that the city contains, imposes the search for a 'simplified' representation [3]. This kind of simplicity is one that does not imply the elimination or reduction of information but rather keeping it visible and enabling the complexity to be deciphered. This is generally performed in urban maps through the translation of complexity into graphic and symbolic systems [4] (p. 153). "The representation of the environment and territory imposes very high levels of abstraction, codification, and normalization, and its practice is therefore inescapable of an interdisciplinary effort and involves an unprecedented mix of different methods of representation, the use of which complements, and is unparalleled in the field of graphic models" [5].

Representations, images, and physical ideas of the land play a primary role in orienting and defining transformation projects and processes. Such images engage discursive practices and establish limits and boundaries with respect to what is expressible, while providing explicit and also implicit rules, as shared, with respect to how the contents of representation interact. Thus, this is both a study of present and past territorial realities and a foreshadowing of their possible futures. Images and ideas of territories have played a decisive role in directing and defining practices of conceptualizing and reconstructing the physical city. There are points of transition and rupture, where new images deconstruct previous ones, paving the way for new meanings and values, while maintaining a character of continuity [6] (p. 36).

## 2. Materials and Methods: A Specific Regulatory Context

The contribution in the following paragraphs dwells on a significant experience, which is quite varied in its application both in terms of the quality and number of experiences involved. The experience of one of the authors in the role of Councillor for Urban Regeneration with the responsibility for urban planning constitutes a privileged point of observation. This makes it possible to witness the constitution of the General Urban Plan (PUG) of the Municipality of Parma that is being realized in this period, triggering reflections precisely on the change in the influence of representation as a technical and programmatic content medium expressed through symbols appropriate to these scales of representation.

Therefore, it addresses a specific case study by discussing the changes in representation tools and graphic languages over time, and then dwells on the digital and interactive technologies necessitated by technological developments and new requirements for the interaction, participation, dissemination, and consultation of urban planning tools.

Since the entry into force of Emilia-Romagna Regional Law No. 24/21 in December 2017: *Disciplina regionale sulla tutela e l'uso del territorio* (regional regulations on the protection and use of land), the processes guiding land development have profoundly changed, in principle, in the structuring through participatory processes and in the way that contents are expressed and represented [7]. In fact, the theme of landscape governance outlined in Law No. 24/2017 involves substantial differences with respect to the tools of the past, starting with the concept of limitation of land use, which must tend towards zero in 2050. The fundamental principles of the law are several.

The aims that are outlined are concepts like the containment of soil consumption as a common asset and non-renewable resource, the regeneration of urbanized territories and the improvement in urban and building quality, and the protection and enhancement of the territory in its environmental and landscape characteristics. In the same way, other questions are requests such as the protection and enhancement of agricultural territories and their agricultural and food production capacities or the contribution to the protection and enhancement of the historical and cultural elements of the regional territory. These principles and others (promotion of greater levels of knowledge of the territory and the existing building heritage to ensure the effectiveness of protection actions and the sustainability of transformation interventions) find value in attributes value to the entire process

of forming the cognitive framework, which becomes the basis on which the actions of the new urban planning instruments are based, and of the strategic lines that structure the General Urban Plans, which must be expressed in form and content through a cartography of an ideogrammatic type.

Regional Law No. 24/2017 expresses itself explicitly on the theme of participation, for which the identification of a special figure of Guarantor (of communication and participation) is envisaged, making it necessary to have numerous moments of comparison and sharing with professional associations, trade associations, and, obviously, citizens, in order to define a tool that can truly be the fruit of an open process of defining the themes, firstly, and then the strategies leading to the city of 2050. The analysis of Law No. 24/2017 and its contents in terms of new planning tools also becomes an opportunity to reflect on the tools through which the governance of the territory is implemented, from GIS to web GIS up to the changes in the graphic language, with particular reference to the planning tools referring to the city of Parma, on which the new General Urban Plan (PUG) is currently being defined.

A map as a device made of signs is capable of creating and conveying meanings that depend on historical and social context. "So, it becomes a product relative to the social context, but also to the interpreter who produces it, apart from the purpose for which it was made" [8] (p. 91). An examination of the contents of urban planning archives makes it possible to identify the various forms through which the dissemination of urban planning tools can take place. It also outlines a condition that is poor and lacking in consultation and dissemination devices in terms of the potential that could be expressed by systematic publication forms of past urban planning archives and through correlation forms between urban planning that has already emerged on digital and old digitalized maps. In urban planning archives, however, the state of the art expresses a very fragmentary condition not only in Italy.

Except isolated experiences, in Italy there is a substantial heterogeneity of tools dedicated to the consultation and digital archiving of the maps of past urban planning, and little use of forms of management, curation, and publication of these documents that can homogeneously affect urban planning heritages. From the analysis of the existing literature in the field of urban archives and the experiences of digital archiving present in our country, we can understand how, in an era in which growing attentions are directed to the practices of 'building on the built', the opportunities offered by the processes of digitization of urban planning documents in urban regeneration practices may prove to be significant [9] (p. 488).

Today, indecision related to urban planning often finds quick solutions in conference rooms or city council chambers. In these venues, it is difficult for administrators and citizens to remember what the urban context really looks like, and often the worst urban elements remain in their memories. In some debates about the urban form, when photographs of the places in question are used, or physical models or cartographies and images are used, memories improve considerably and the choice assumes reasonableness and depth. Speaking of archives, the reference is a very articulated network for which, across international, national, and local contexts, it is not always possible to identify and define the characteristics of uniformity. In Italian municipalities, there are few virtuous realities that have started the difficult and costly process of digitizing planning archives. Among them, the experience of the Municipality of Bologna and the Municipality of Modena stand out, joined in more recent years (2011) by the Municipality of Parma. Through this source, it has been possible to carry out an examination of the plans and their peculiarities of thematic representations [10].

For all these reasons, this paper focuses on a practical example of the application of Emilia-Romagna Regional Law No. 24/2017. As it is a regional law, it applies only to the Municipalities (or Unions of Municipalities) of this Italian region which are drafting urban planning tools according to these principles that lead to the drafting of plans that are not

confirmative but that identify strategic lines of development and growth of cities based on visions and actions aimed at implementing the strategies.

*Representation of Parma's Planning: From the PRG of 1887 to the PSC2023 Approved in 2019*

The first urban development plan drawn up for the city of Parma, dated 1887 (Figure 1), describes four basic concepts at a scale of 1:2500, using just four colours: built-up area (in grey, indicating the already built-up area), developable area (in red, indicating expansion areas), demolitions (in yellow, indicating above all the areas of redefinition of the urban fabric through the demolition of buildings or infrastructures), gardens, vegetable gardens, lawns (in green, indicating existing green areas). This first tool is therefore limited to indicate the transformation of the city, most of which has not been implemented, especially in terms of planned demolitions. As is evident from the sample images below compared to the past, elaborate scrolls and frames are disappearing in favour of a more essential representation with an increasingly communicative intent, which is often a synthesis of images from different scales and explanatory legends, or statistical data. In this way, the person that reads a map is able to translate a graphic expression into a linguistic expression thanks in part to the legend that represents the point of conjunction between the two forms of expression [8] (p. 94).

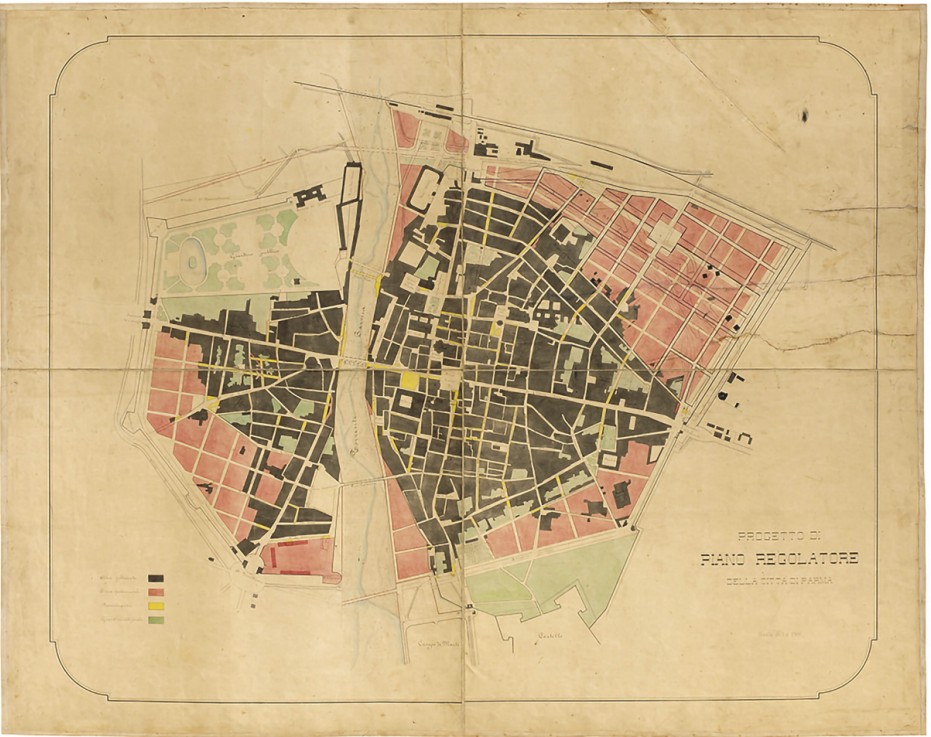

**Figure 1.** Master plan project for the city of Parma, 1887, original scale 1:2500.

The second town plan, dated 1910 (Figure 2), is a simple extension plan. Here, with extremely basic graphics, on the basic cartography at a scale of 1:2000, the colour black is used for the representation of the existing fabric, with the colour red simply outlining the urban layout of the planned extensions in the northern and southern parts of the city, without giving any kind of functional or typological indication, either of the actual or planned state; only a few of the infrastructural lines drawn here had actually been built (e.g., the current Viale Milazzo).

The later 1938 plan (Figure 3) was drawn at a scale of 1:2000 and is much more clearly defined and articulated than the earlier plans. Red lines are used to indicate the directions of the redesigned city layout, highlighting a series of "extensive areas" of residential character ("new urban blocks") around the city, except for the northern area, which follows a line that

was already established by the demolition of the walls, and which, because of its proximity to the railroad, is designated for the location of industrial buildings. A detailed legend explains the numbers on the map, which partly define the zones and partly describe their transformation lines. There is also a second legend, which is very interesting in terms of contents. This legend describes both the characteristics of what already exists (e.g., in green: existing green zones) and what the plan provides for (e.g., in green with internal dotting and reinforced border in dark green: new green zones). In this legend, an articulation of content is anticipated through the use of different colours, screens, borders, and line types that will find wide application in post-war plans.

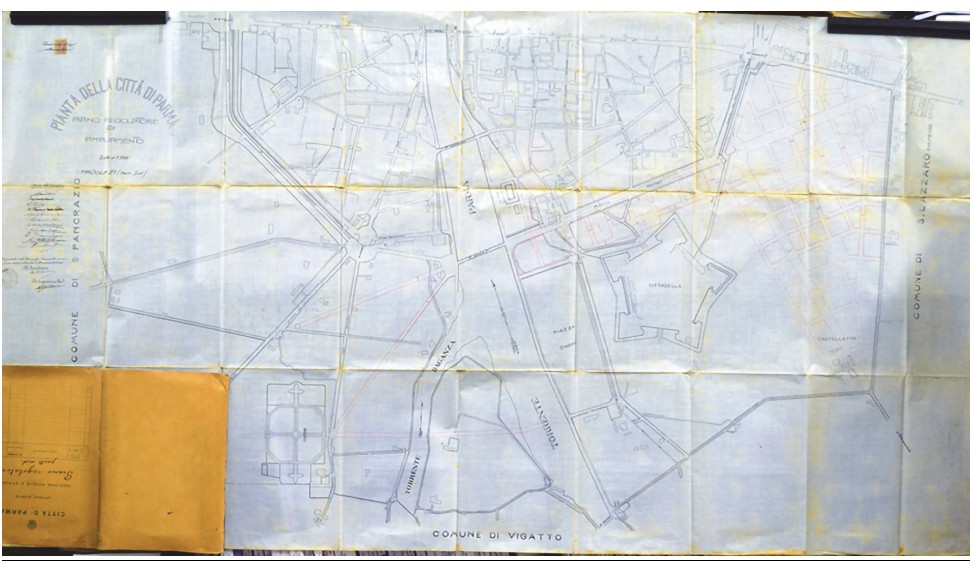

**Figure 2.** Extension master plan for the city of Parma, 1910. Board 1 (southern part). Original scale 1:2000.

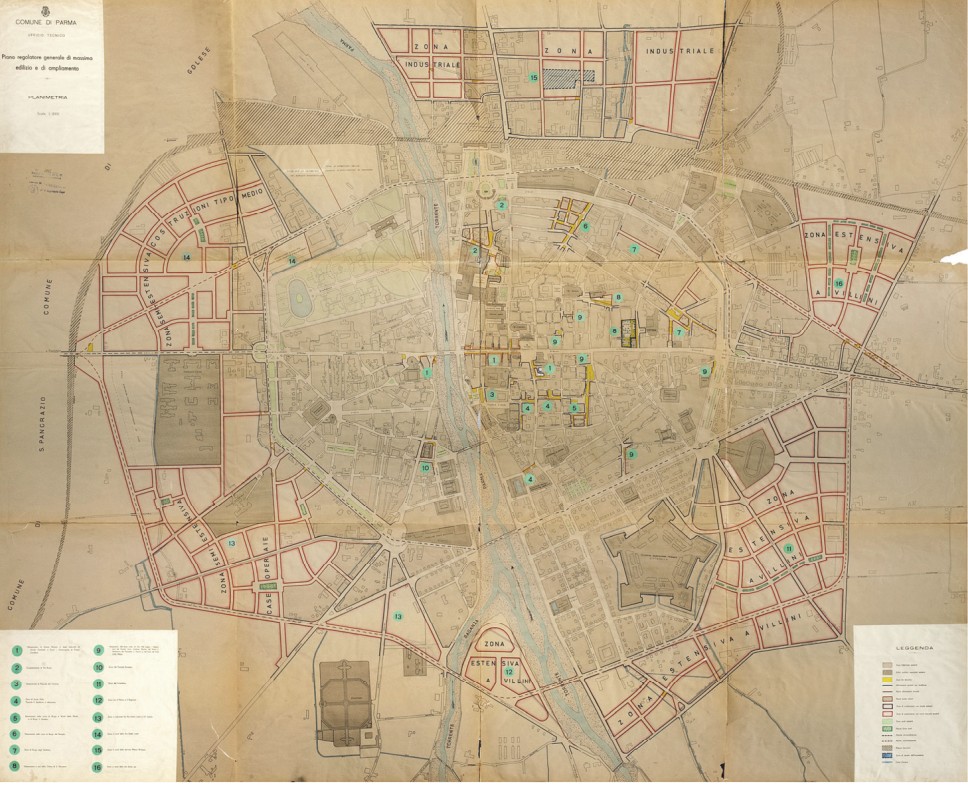

**Figure 3.** General construction and expansion master plan for the city of Parma, 1938. Original scale 1:2000.

The 1950 plan (Figure 4) at a scale of 1:5000 is in fact a "reconstruction plan of the war damaged city centre" and not a true development plan of the city, as in the legend it is highlighted through only three colours denoting the "state of the built-up area as a result of the damage suffered", distinguishing the lightly damaged buildings (in yellow), the badly damaged buildings (in orange), and the destroyed buildings (in brown). In fact, an overview of damaged buildings is included in the elaboration, distinguishing and summarily quantifying the extent of the damage. The colour is applied to a heliographic reproduction that represents for the first time a basic cartography that follows a model similar to the Regional Technical Cartographies (CTR).

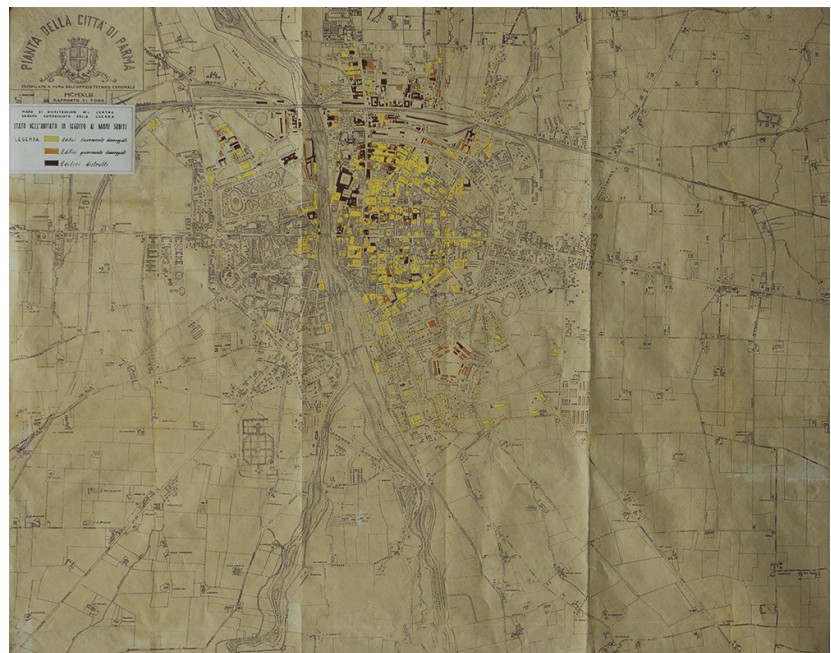

**Figure 4.** Reconstruction plan of the war-damage of the urban centre, 1950. Original scale 1:5000.

The General Regulatory Plan (PRG) of 1963 (Figure 5) drawn up at a scale of 1:5000 for the first time distinguishes the zones of the historic centre from those parts of the city outside it. The representation used, well described in the legend, separates for each functional category the existing from the planned, using colours, lines, hatches, and symbols that provide a very rich and detailed description of the existing and planned urban fabric in all its different functional and infrastructural articulations.

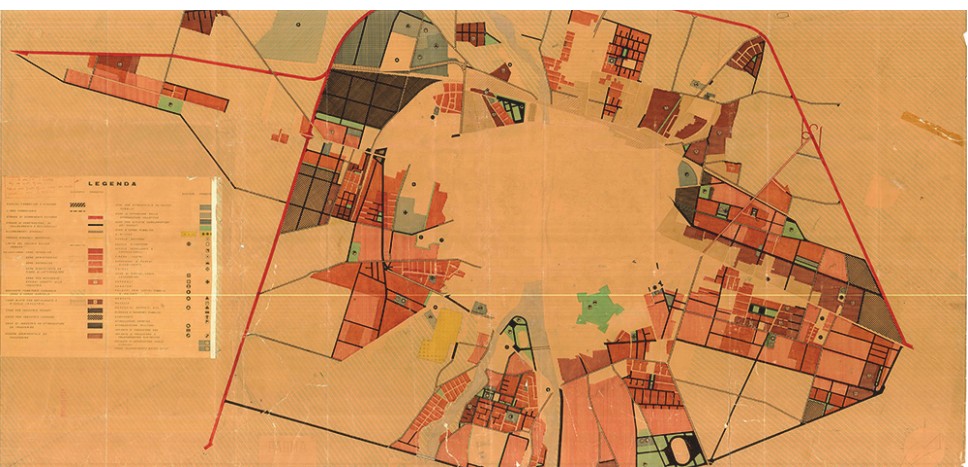

**Figure 5.** General Regulatory Plan (PRG) for the city of Parma, 1963. Original scale 1:5000.

Of this plan, there is an overview board at a scale of 1:25,000 (Figure 6) that frames the territory of the entire municipality and in which appears for the first time the beltway, modified several times over the years. Also in this board, drawn up using the representation of the Military Geographical Institute (IGM), it is defined in a concise way as a base map with the legend that highlights through different colours and hatches the functional zones of expansion of the city.

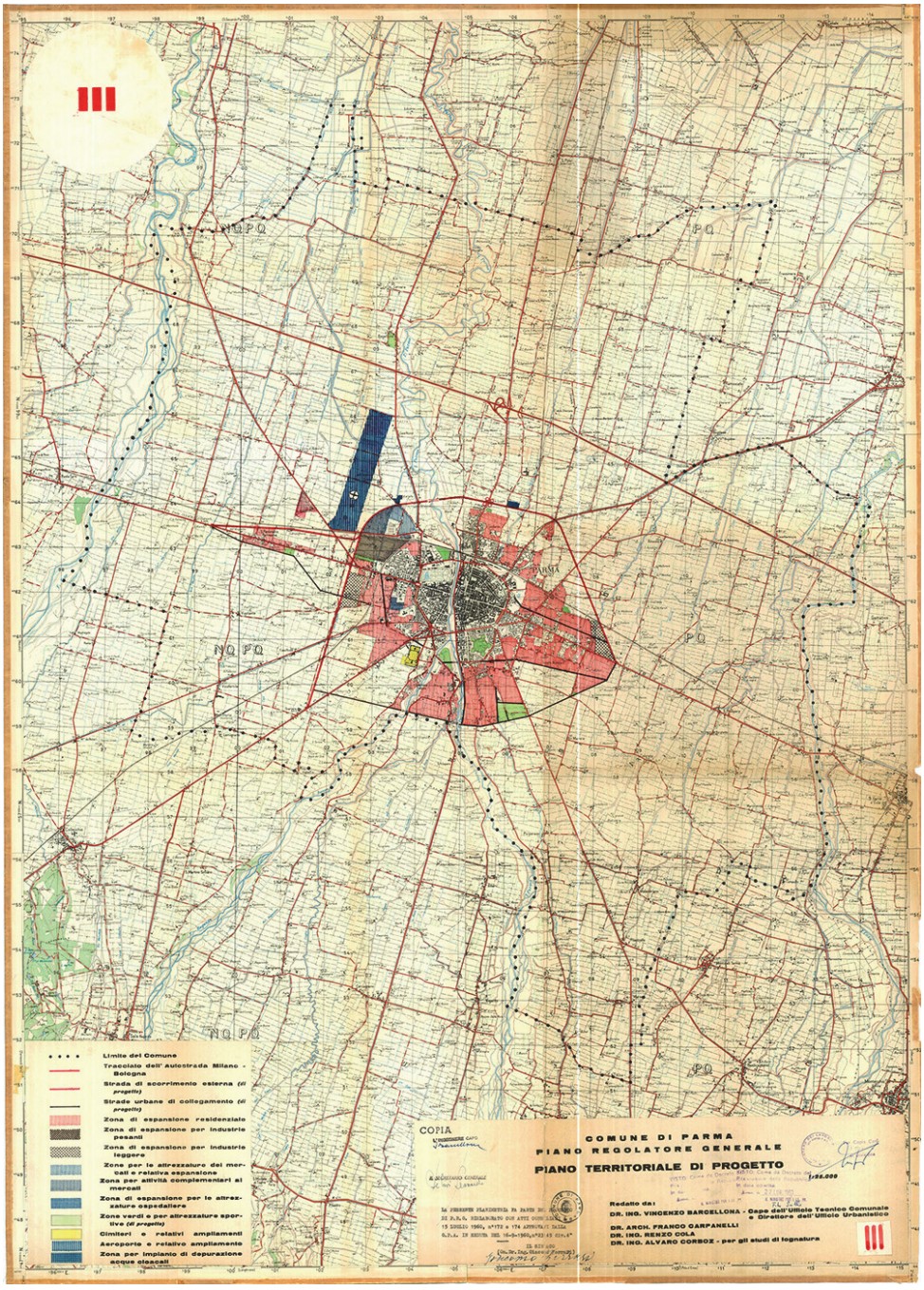

**Figure 6.** General Regulatory Plan (PRG) for the city of Parma, 1963. Original scale 1:25,000.

The General Regulatory Plan (PRG) of 1974 (Figure 7), drafted at a scale of 1:5000, consolidates the use of screens to express functional zoning about the areas outside the historic centre. The support here is a tracing paper on which work is conducted by ink and screens having different hatchings and grayscales, which through the process of holophote reproduction allows the plans to be replicated and diffused. Using black and

white and grayscale, the representation is definitely less immediate and the meanings of the different types of functional zoning are described through very accurate and articulate legends (Figure 8).

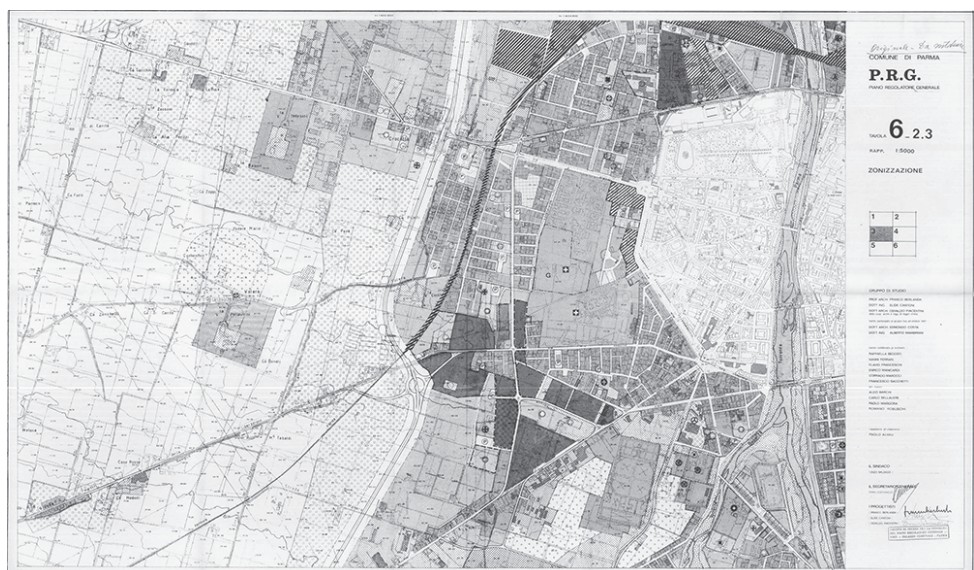

**Figure 7.** General Regulatory Plan (PRG) for the city of Parma, 1974. Board No. 6.2.3. Original scale 1:5000.

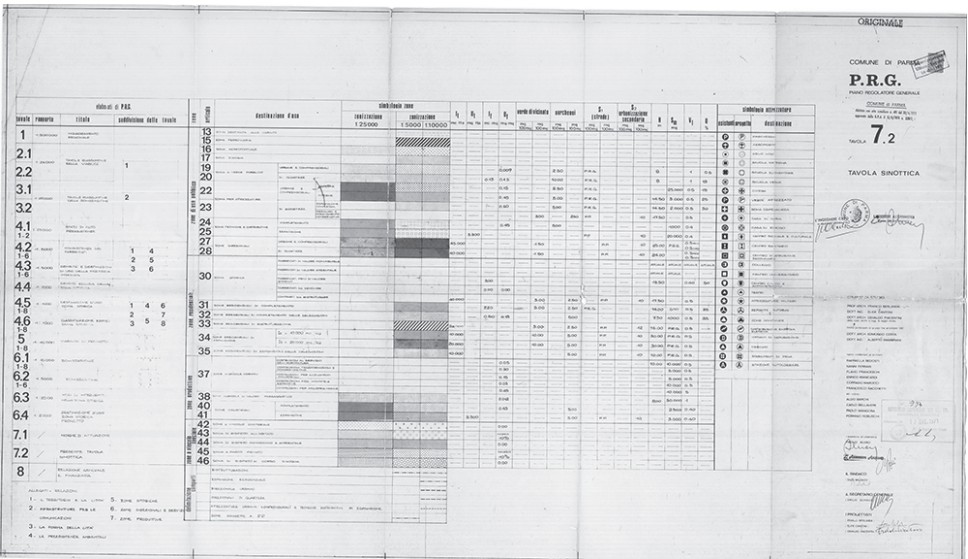

**Figure 8.** General Regulatory Plan (PRG) for the city of Parma, 1974. Synoptic Board. Original scale 1:5000.

The variant of the PRG (1978) follows exactly the same lines of representation of the functional zoning (Figures 9 and 10); just as similar, but with a much less articulated legend, are the boards of the 1980 "Detailed Specification for Interventions in the Historic Center", in which, again with screens and borders, not the functions but the categories of intervention planned on the different buildings of the historic fabric are indicated (Figure 11).

The General Regulatory Plan (PRG) of 1998 (Figure 13), at a scale of 1:25,000, introduces an important innovation in the representation/communication of functional zoning, that is the use of colour, which by now modern reproduction systems allow to be used making the recognizability of the different categories that are going to be described on the different portions of the municipal territory much more effective.

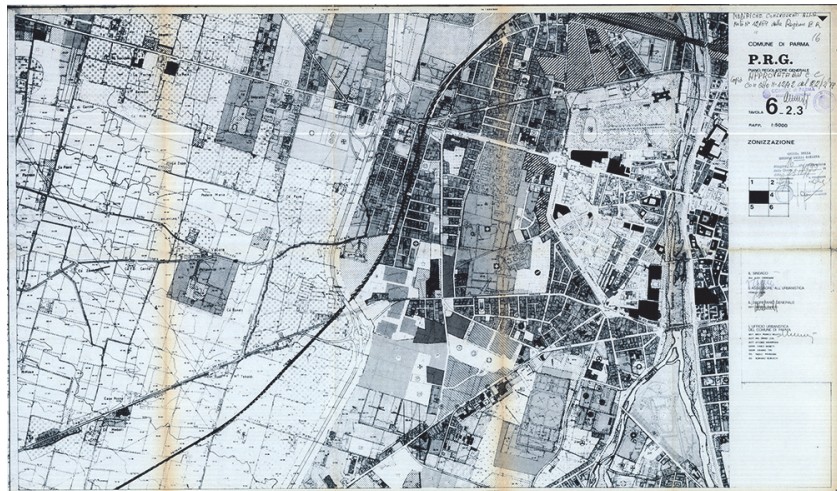

**Figure 9.** Variant of the General Regulatory Plan (PRG) for the city of Parma, 1978. Board No. 6.2.3. Original scale 1:5000.

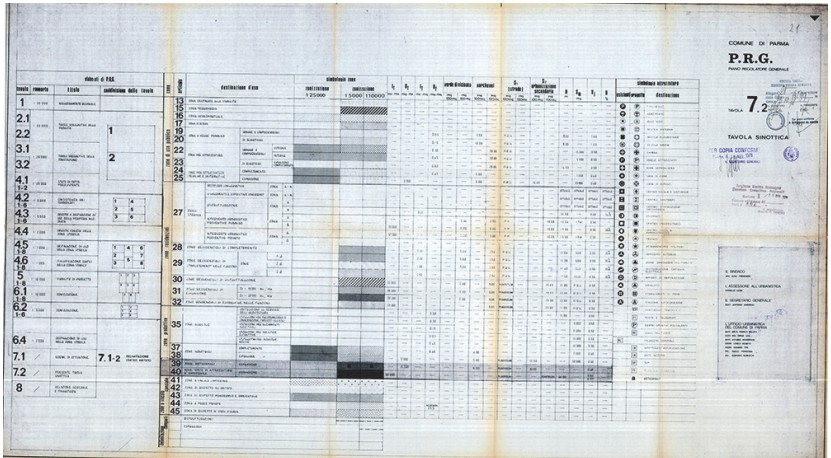

**Figure 10.** Variant of the General Regulatory Plan (PRG) for the city of Parma, 1978. Synoptic Board. Original scale 1:5000.

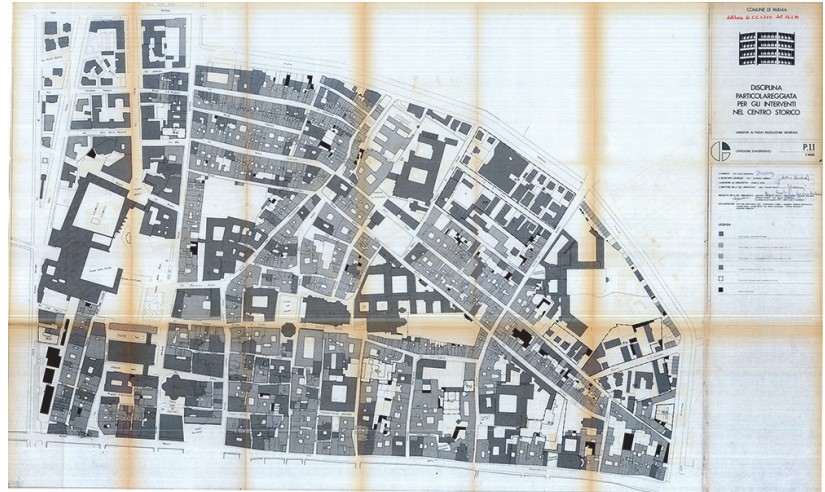

**Figure 11.** Detailed regulation for intervention of the Historic Centre of Parma, 1981. Board P.1.1- Intervention's categories. Original scale 1:1000.

The variant of 1989 also keeps to the same graphic modes and lines as the previous one of 1978 (Figure 12a,b), although it expresses the meanings of the various screens in a more concise legend than the previous one.

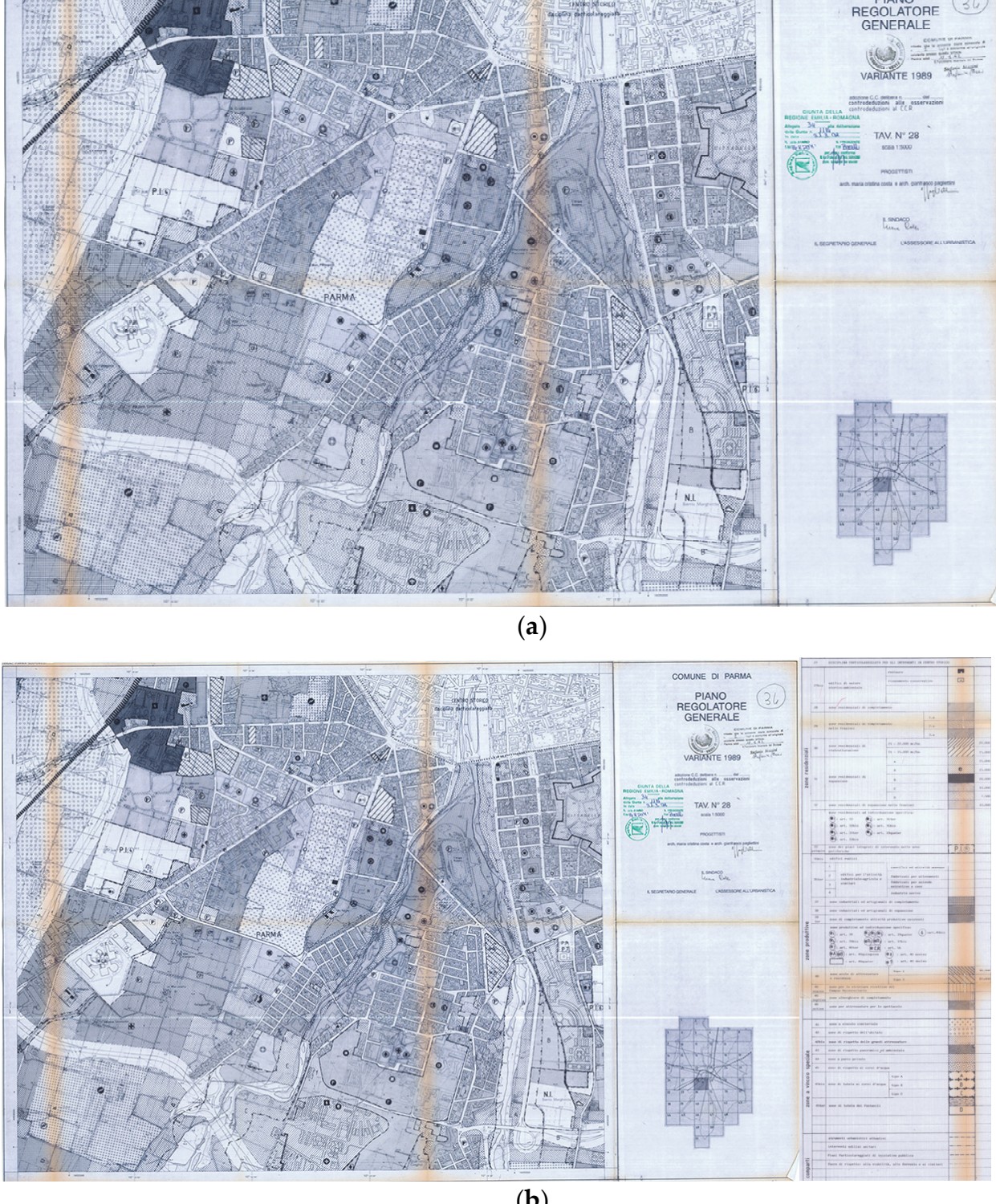

**(a)**

**(b)**

**Figure 12.** (**a**) Variant of the General Regulatory Plan of 1974 (PRG) for the city of Parma. Board 28, 1989. Original scale 1:5000. (**b**) Variant of the General Regulatory Plan (PRG) for the city of Parma. Synoptic Board, 1989.

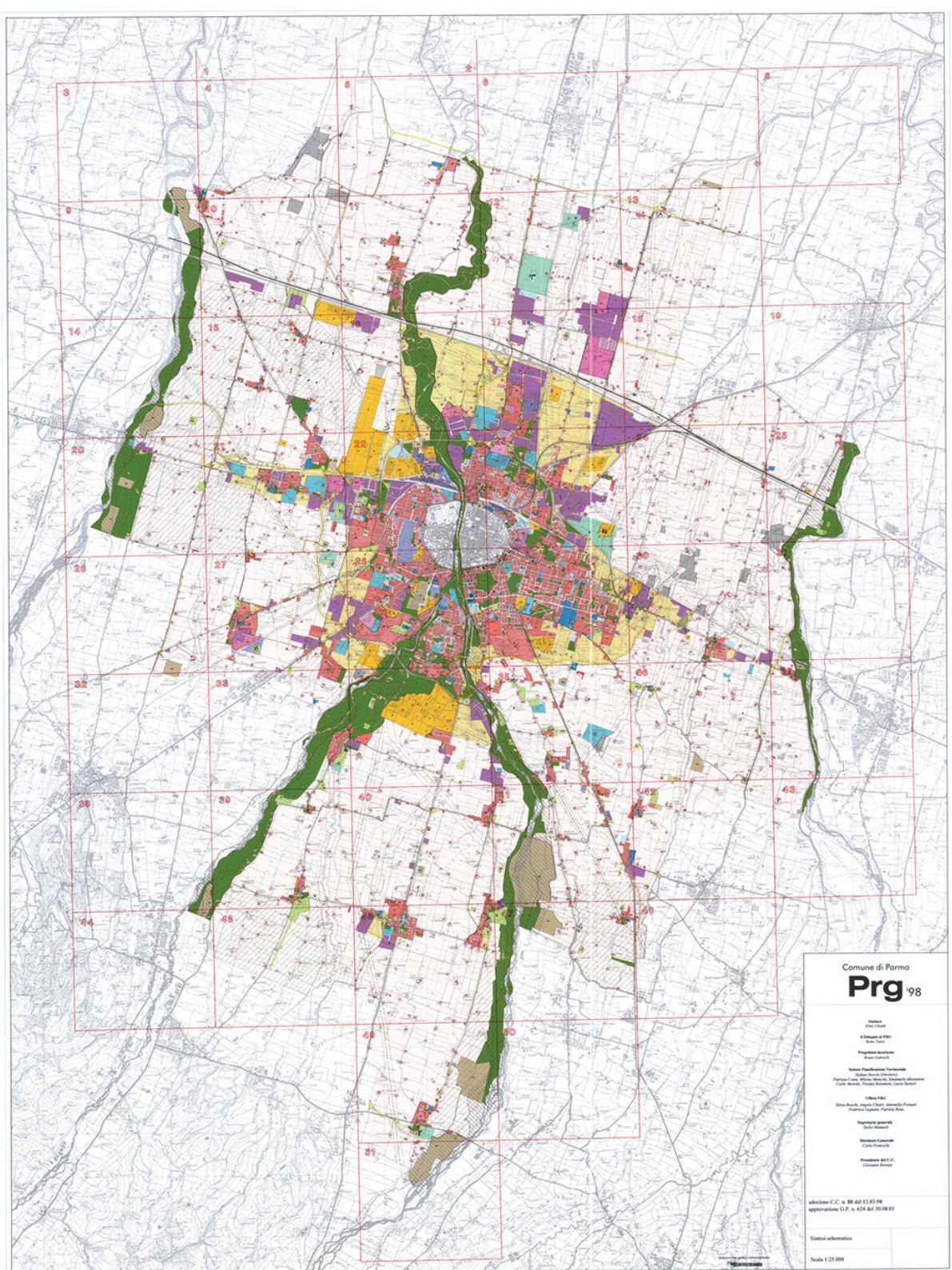

**Figure 13.** General Regulatory Plan (PRG) for the city of Parma, 1998. Schematic Board. Original scale 1:25,000.

Along the same lines are the Municipal Structural Plan and Municipal Operational Plan of 2008 (Figure 14), in scales of 1:25,000 and 1:10,000; the later contribution of the Urban

Building Regulations (2010), whose cartography is divided into A3 sheets at a scale of 1:2000, was grouped in 51 files corresponding to the 51 panels of the previous cartography.

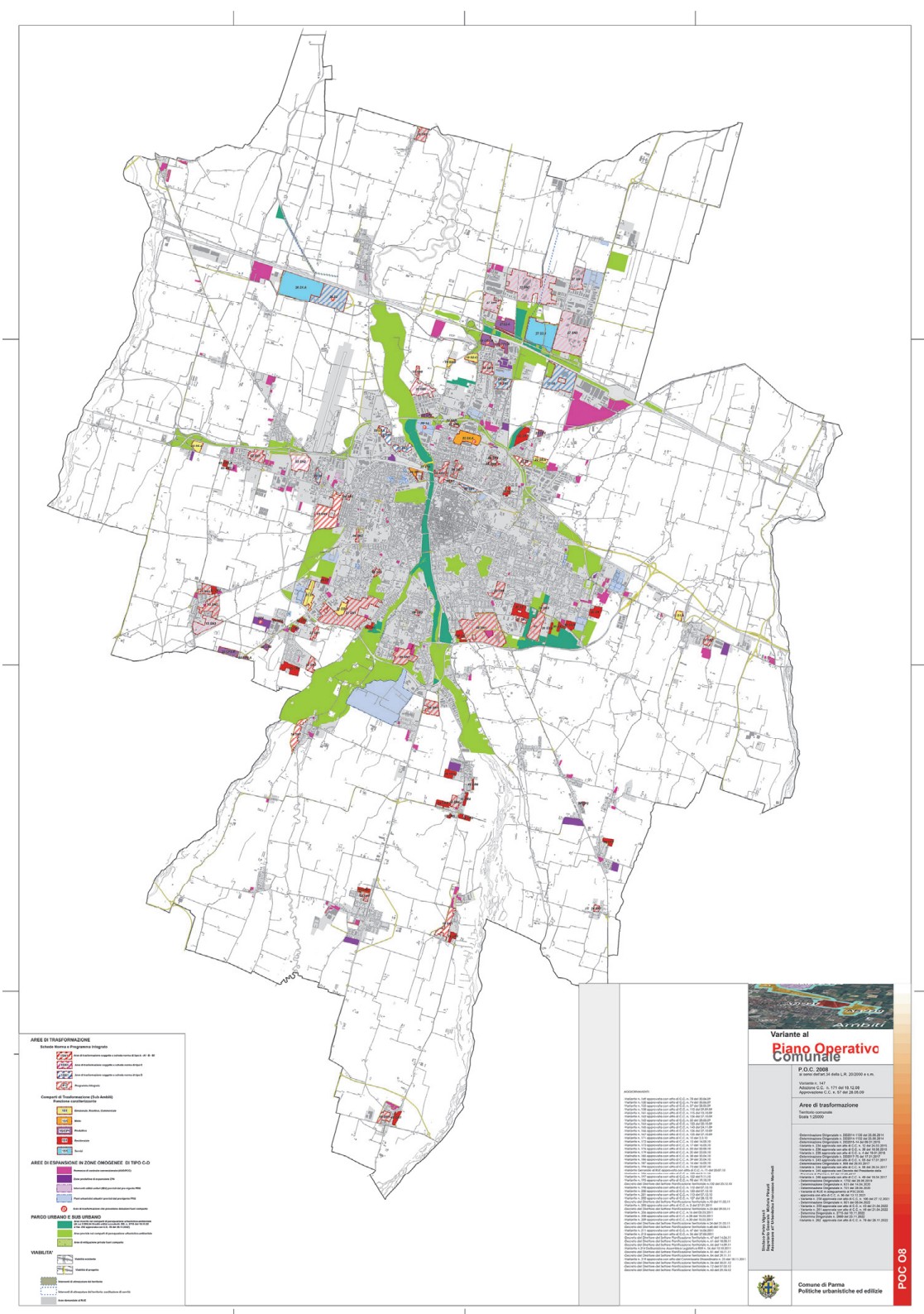

**Figure 14.** Variation in the municipal operational plan of 2008. Transformation areas. Original scale 1:25,000.

In the latest variant of 2021, the Urban Planning Regulations (RUE) is expressed at a scale of 1:5000 (Figures 15 and 16), always using overlapping borders, colours, and screens to express land planning and development.

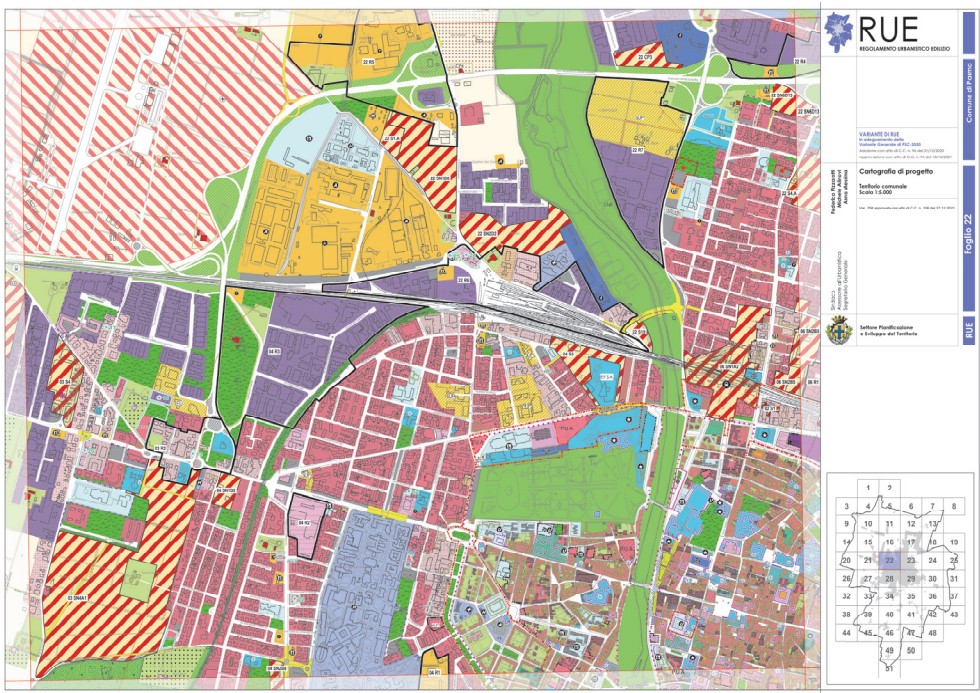

**Figure 15.** Variation in the Urban Planning Regulations (RUE) in adaptation to the general variation in PSC 2030. Sheet 22. Original scale 1:5000.

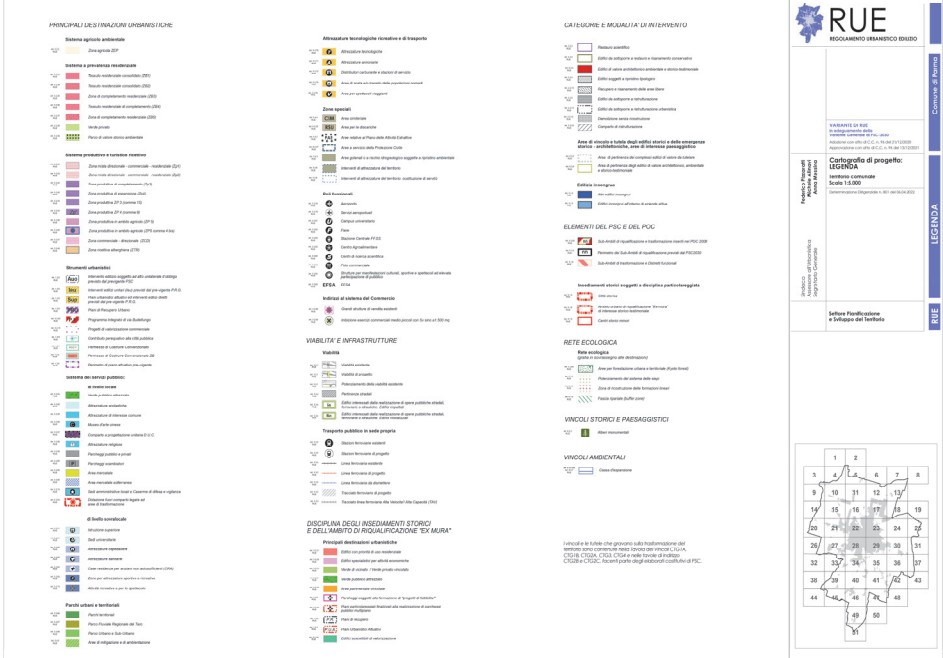

**Figure 16.** Variation in the Urban Planning Regulations (RUE) in adaptation to the general variation in PSC 2030. Project cartography. Legend. Original scale 1:5000.

The variant to the Municipal Structural Plan, a general urban planning tool relating to the entire municipal territory, to outline strategic choices, which came into effect in 2019 (with the Municipal Structural Plan, named PSC2030) is expressed at a scale of 1:25,000

and 1:10,000 and once again uses borders, screens, and colours (Figure 17) to express the urban planning policies planned for the territory, highlighting the different types of consolidated cities, transformation, functional districts, and rural territory to identify the lines of transformation.

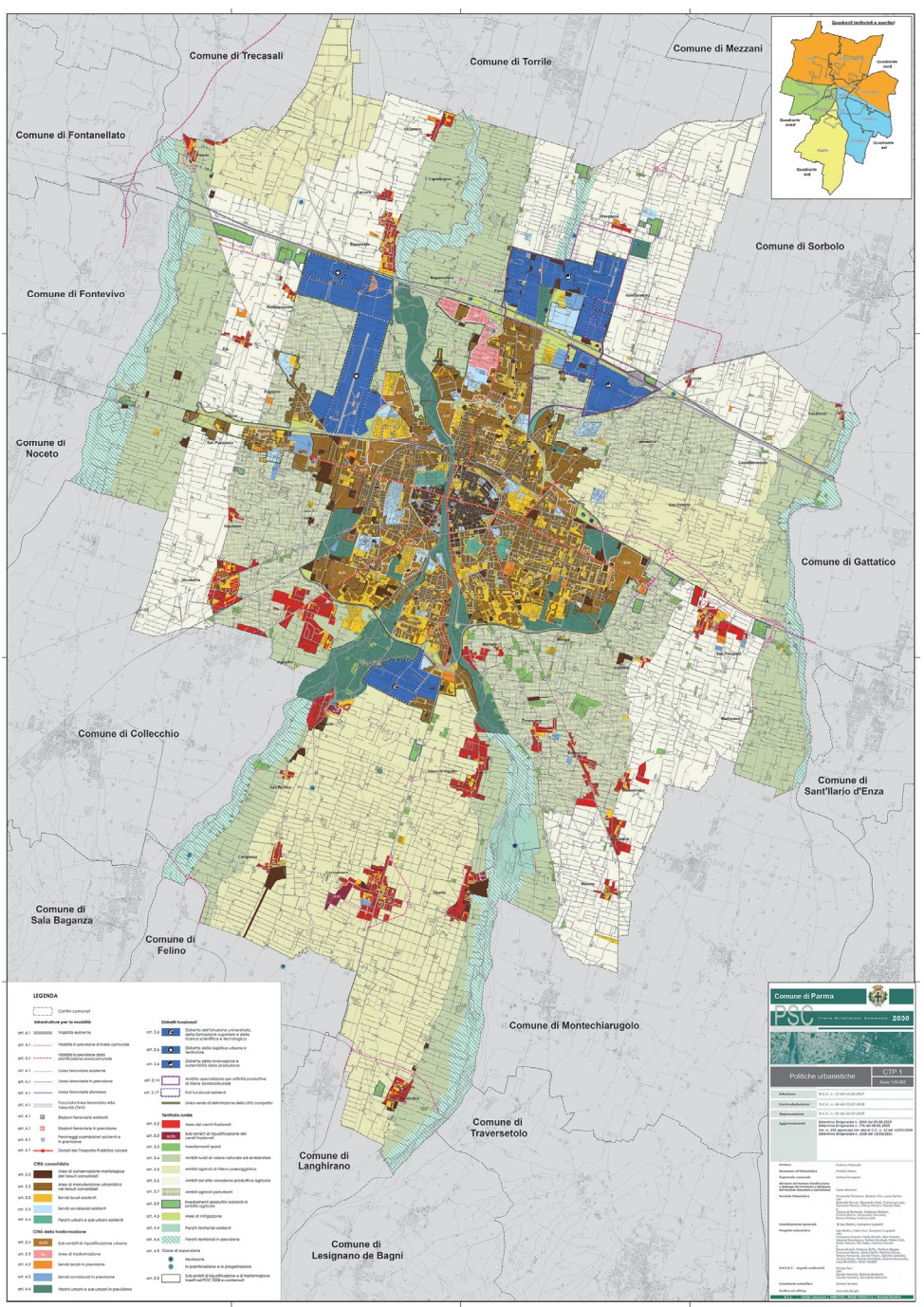

**Figure 17.** Data from 2019, Municipal Structural Plan 2030. Town planning policies CTP1. Original scale 1:25,000.

The long excursus was carried out by reading the plan cartography of the city of Parma from its first manifestations to the most recent expressions. This excursus allows us to make some reflections on several levels. On the one hand, the theme of signs used in planning cartography, which is usually expressed at scales ranging from 1:25,000 in the case of summary maps up to a scale of 1:10,000 (only in the historical centre), requires signs to be used in a strongly symbolic way, to build a specific language for initiates who know

the interpretations that lead to the correct interpretation of the contents, thanks to very articulated legends and synoptic tables which, both in the black and white and in the colour versions, contain precisely the heart of the development of the city.

## 3. The Representation of Parma's Planning in PUG PR050

On 21 December 2017, the new Emilia-Romagna Regional Planning Law No. 24/2017 "Regional regulations on the protection and use of land" was approved and it came into force on 1 January 2018. With this law, the Emilia-Romagna region has equipped itself with new tools oriented toward increasing the attractiveness and liveability of cities by activating urban regeneration policies, containing land consumption, and increasing the competitiveness of the regional system. In order to simplify municipal urban planning and enhance negotiation processes in defining the operational phase of interventions, municipal urban planning according to the new regional law is divided into:

(a) A single General Urban Plan (PUG), which establishes the discipline of municipal competence on the use and transformation of the territory, with regard to the processes of reuse and urban regeneration;

(b) Operational agreements and an implementation plan of public initiative with which, in accordance with the PUG, the municipal administration allocates building rights, establishes the detailed regulation of transformations, and defines their contribution to the realization of the objectives established by the strategy for urban and ecological–environmental quality.

In particular, the PUG is the planning tool that the municipality formulates, with reference to its entire territory, in order to outline the structural invariances and strategic choices of urban planning and development within its competence, oriented primarily towards the regeneration of urbanized territory, the reduction in land consumption, and the environmental and territorial sustainability of uses and transformations [11].

The city of Parma started the drafting of its General Urban Plan (PUG), with a contract signed on 5 May 2021, the *Raggruppamento Temporaneo di Imprese* (R.T.I.), whose mandatary partner is the firm UNLAB (Urban Landscape Architecture Bureau) coordinated by architect Andreas Faoro.

### 3.1. The Levels of PUG PR050

The structure of the plan is organized on five main levels. The first level is made up of the themes that describe the city and the territory, providing the basis for comparison between the various actors. The second level constitutes the visions declined by 10 "representative images" that establish the objectives of the plan in the medium and long term as a result of the intersection of the strategic objectives (defined by the EU and UN agendas) with the themes that emerged from the participatory project together with the cognitive insights and demographic and socioeconomic surveys.

The third level are the scenarios or rather the strategies of the plan represented by seven 'figures' of transformation. They are supported by actions, prioritized and necessary, which act in the short, medium, and long term and contribute to achieving the objectives. The actions and scenarios define the operational character of the strategy and at the same time are able to verify the consequences of choices that can be made in view of certain objectives.

The fourth level includes the places of transformation and strategic projects, or atlas. It constructs the urban geography of Parma 2050 and identifies the parts of the city that for their importance can act strategically in the realization of the vision. Through their design, the spaces identified can make new contributions in terms of programs, functions, and innovative infrastructures at the service of the city as a whole.

The ten images of the vision, therefore, indicate a direction for Parma on the 2050 horizon: a city of biodiversity, a polycentric city and neighbourhoods city, a city of shared and sustainable mobility, a city of culture and widespread knowledge, a city of quality living and care for people, a city of development and opportunities, a city of agriculture as

historical-environmental and socio-cultural heritage, a city capital of food and sustainable food, a city of renewable energy, an inclusive city, and a city of wellbeing.

These define the fundamental orientations, guide the choices of the different parts of the territory and, even if these are not fully realized in 2050, indicate the path to follow in order to tackle the current problems and challenges ahead.

The scenarios specify the strategies and make the objectives contained in the 'pictures' more concrete. The 'pictures' are supported by a series of actions, which 'inform' the strategies to realize the objectives and achieve the results within the set timeframe. The list of actions is an open list and susceptible to change, in the sense that the actions can be added but not deleted, that its priorities will sometimes have to be reprioritized, but it is certainly an inclusive list that can be reinforced along the way with more detail.

The identified actions, consequently, develop some scenarios, in order to realize the customary objectives. They are the groundwork for the actions in the short, medium, and long term. These cross the different policies of the city, as an outcome deriving from the themes shared with the participatory project and included in the "charter" which represents the fifth level of the plan that includes the discipline and rules to implement the strategic projects and the future building regulations.

From welfare (health, sport, education), to town planning (redevelopment and urban regeneration), public works (public space, road infrastructure, energy...), and the environment and landscape (conservation, preservation...), the actions accompany the implementation of the vision and are the operational basis of the strategies.

The strategic spaces define specific and fundamental areas for the realization of the vision and are representative of the strategic themes related to transformation. Their transversal character makes them aspects to be investigated with greater precision, to be investigated in a design sense as complex operations of transformation.

Within each strategic space, a number of projects can be recognized, i.e., areas in which it is possible to identify the actors involved, establish the main programmatic and morphological structuring lines, and the timeframe of realization. These projects often enjoy high visibility and must therefore be conceived as exemplary, prototypical interventions that can play a driving role in the construction of the Parma of the 21st century: concrete realizations capable of measuring the ambitions of the New Parma.

Strategic projects are flanked by pilot projects that have an experimental character, as they can act in the short term and do not require significant economic resources. In addition to these are projects that fulfil generic needs, called 'general projects'.

The whole PUG allows progress in the direction indicated by the vision. As a whole, the PUG is represented in a series of tables starting from those organized in the "Knowledge Framework", to the ideogrammatic one of the "Structural Elements", and the strategic planning scheme.

The PUG as far as 'discipline' is defined at a scale of 1:5000, it collects and represents spatially the ordinary measures of transformation and homogeneous tissues; these panels cover the entire territory but with different 'grains' they open like a hypertext to in-depth studies or diagrams at a closer scale as in the case of the historic centre.

*3.2. The Strategics Maps of the PUG of Parma*

The strategic maps at 1:25,000 and 1:10,000 scales are more selective. In fact, the strategic layout plan, as well as the map of the places of strategic transformations say that there are spaces that should be transformed for the benefit not only of the narrow context in which they are located but of the entire city. The plan brings with it some design proposals together with the representation of the new strategic structural elements of the new spatial configuration of the city of Parma that includes the so-called Unique Text (TU) and the communities dispersed throughout the territory. It should be recalled that the strategic planning scheme refers in a priority manner to the transformations of a general character applicable to the various tissues, while the map of the places of transformations indicates in a more selective manner the areas of superordinate intervention.

The cards contained in the atlas are subdivided by neighbourhood (scale 1:10,000) and hamlets (1:5000) and contain the projects with their respective guidelines with reference to mobility, public space, services, mixing, functional aggregation and transformation of the fabric, and ecological and energy infrastructures consistent with Vision Parma 2050.

Each sheet is conceived as the starting point for more detailed reflections that are realized by reviewing the specific contents and address urban and infrastructural regeneration, requalification, and reorganization in an integrated manner. The sheets set out guidelines and related actions for the urban and environmental quality that each intervention should bring to the specific neighbourhood or area; examples include oil protection, biodiversity, energy (the zero-$CO_2$ transition project by 2050), the strengthening of urban and supra-urban centralities and collective and diffuse habitability, the recovery of the quality of the built environment, and the regeneration of ancient and modern building fabrics.

The representative cartographic elaborations of the PUG-PR050 have ideogrammatic value when these refer to the strategic contents, as indicated in Article 24 of L.R. 24/2017; for this reason, the richness of the contents related to the themes dealt with in the new planning risks being debased by a representation that, although interesting in the new relationship established between signifier and signified of the signs adopted in the cartography at a scale of 1:15,000 (for the urbanized territory) (Figure 18) or 1:25,000 for (the entire municipality) (Figures 19 and 20) at which the plan is expressed, does not render to the utmost the peculiarity and characterization of the strategic projects that must qualify the city of 2050, in all its different meanings of planning scale, ranging from the most far-reaching and large-scale to the most minute projects of even small areas scattered throughout the urban territory.

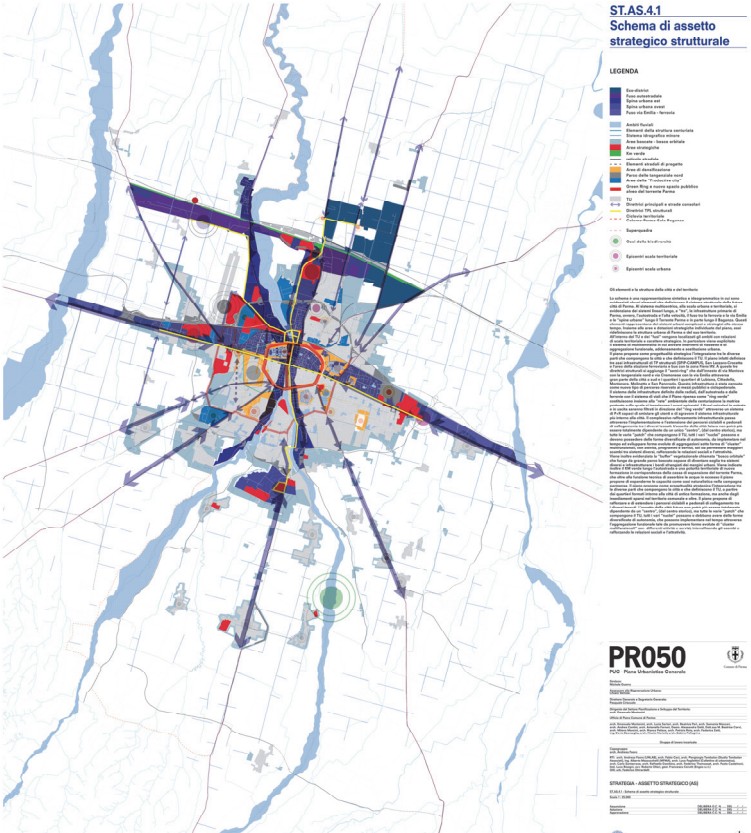

**Figure 18.** Data from 2023, General Urban Plan PR050. BoardST.AS.4.1. Strategic Structural Layout Scheme. Original scale 1:25,000.

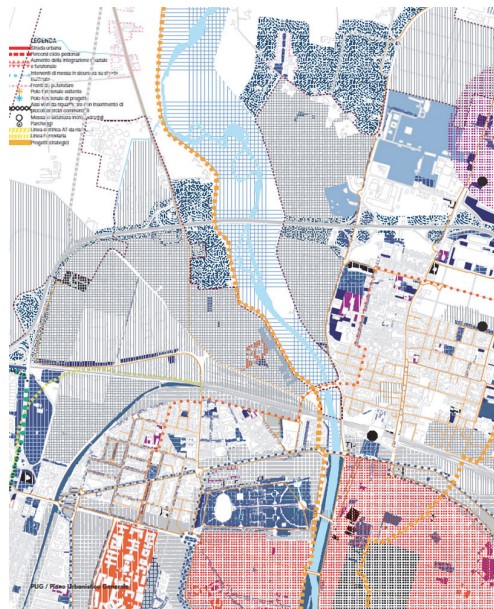

**Figure 19.** Data from 2023, General Urban Plan PR050. Extract from the atlas; north/north-west area. Original scale 1:5000.

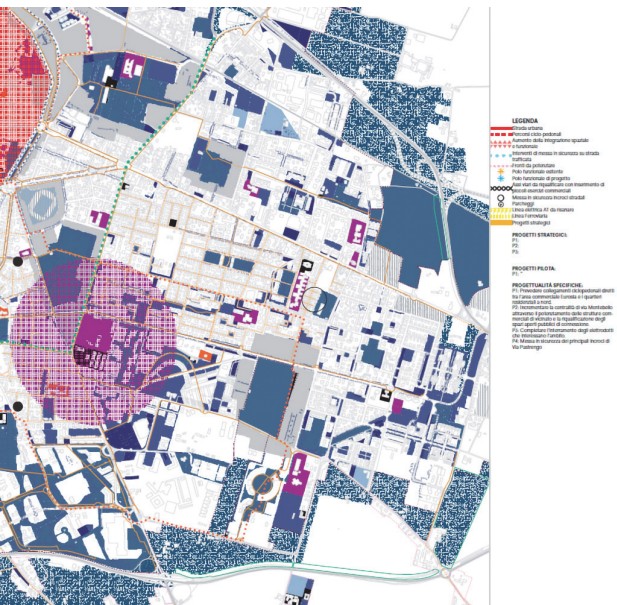

**Figure 20.** Data from 2023, General Urban Plan PR050. Extract from the atlas; east/south-east area. Original scale 1:5000.

### 3.3. The Participative Process in the Formation of the PUG of Parma

The formation of the PUG of Parma began in the summer of 2021 and saw one of the most significant moments in the formation of the technical tables that in July led to the identification of the themes characterizing the city and the territory of the city of Parma. The involvement and participation of different actors provides the possibility of changing the perception and understanding of the phenomena examined [12] (pp. 28–32) and allows the construction of knowledge in support of the plan that is not only correct and relevant but also accepted by stakeholders [13].

A little more than a year later, in September 2022, a first moment of restitution to the city of what had been elaborated by the grouping of planners was carried out, at the *Complesso Monumentale di San Paolo* (Figure 21). All of the stakeholders were invited the

year before to the start-up phase. Over the course of three days—divided into seven thematic tables corresponding to the scenarios that emerged after the "Diagnostic Knowledge Framework"—the contents of this first part of the plan proposal were defined. These proposals were shared and then presented at the end of October 2022 (Figure 22), in the various neighbourhoods of the city. In January 2023, three meetings of the Council Committees on urban planning, productive activities, and civil protection were held specifically addressing the issues of welfare, ecological, and environmental endowments, and the changes that the new tool will bring to the housing market, specifically involving the sector's operators.

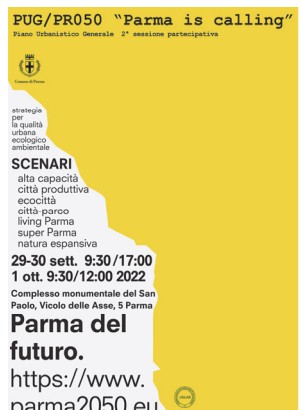 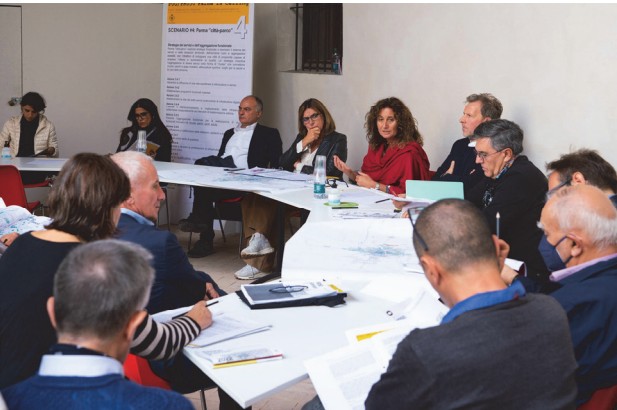

**Figure 21.** Poster "Parma is calling", second participatory session, September 2022.

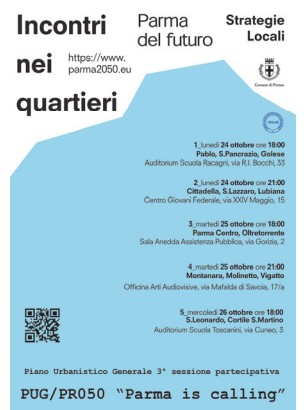 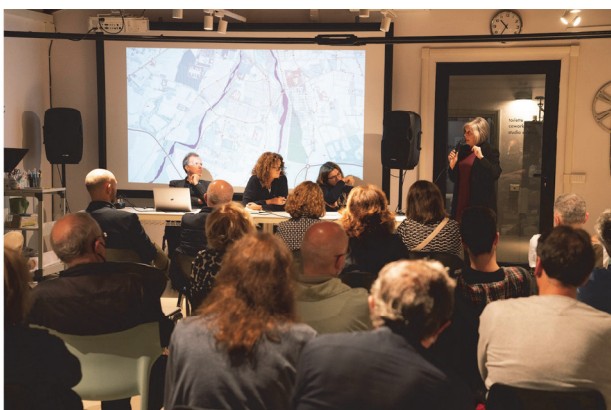

**Figure 22.** Poster "Parma is calling", third participatory session, October 2022.

Numerous technical tables have been implemented during this year, including one dedicated to green infrastructure and ecological–environmental legacies, held in November 2022, and aimed at specific operators in the sector, whose involvement is valuable and decisive for the success of the plan. In addition to these, there have been numerous opportunities for discussion both with representatives of professional associations and trade associations and with representatives of other cities in Emilia-Romagna (in particular Reggio Emilia and Piacenza, neighbouring capital cities) that, like Parma, are grappling with the drafting of the PUG or have even already taken it on, adopted, or approved it.

Finally, now approaching the assumption, scheduled for mid-July 2023, two public presentations were made on 29 and 30 June 2023 (Figure 23) that were useful to introduce the contents of the PUG and to give guidance on how to read the complex documents that make up the regulatory and cartographic corpus, which is decidedly different from that of previous urban planning devices.

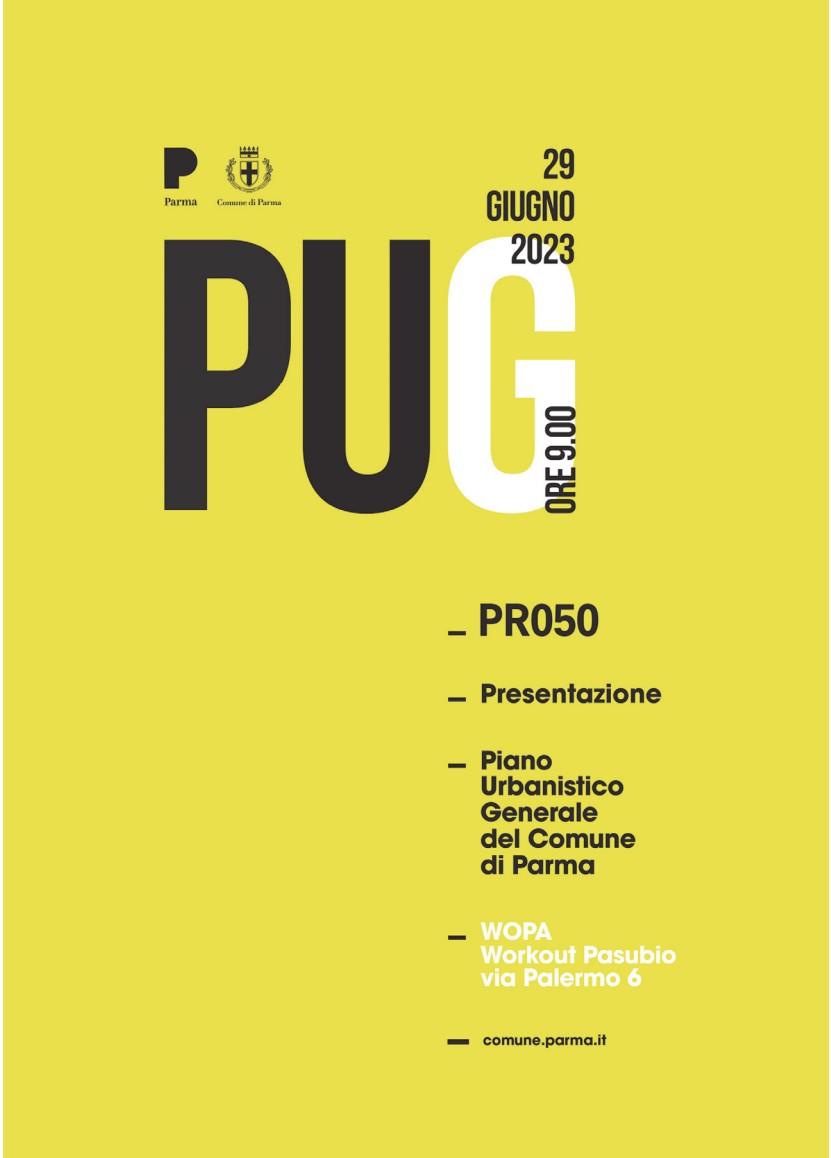

**Figure 23.** Poster of the presentation of the PUG to the citizenship in view of the recruitment, June 2023.

Certainly, the dialogic and participatory functions, and the composition of the archive by specific sections and projects can lend versatility to the infrastructure of publishing urban planning tools by making the digital urban archive an enhanced system of critical reading in the field of urban planning. It also influences, in the local context, the formation of a widespread awareness of conservation issues and the construction of a historiography dedicated to the history of planning [9] (p. 488).

The period following the time of the assumption is to looking for the possibility of submitting comments by technicians and, more generally, by citizens. During that period, the planning office of the city of Parma will open a dedicated information desk, in order to assist and accompany in the reading of the PUG and in the submission of requests. Throughout the duration, public meetings and technical tables will also be held to make the participatory process even more effective in such a crucial moment as the one in which the plan proposal is shared and it is still possible to affect it, thanks to everyone's contribution, refining its contents, before the final steps that will lead to its adoption, at first, and then its approval.

## 4. Discussion: Interactivity with Strategic Maps and WebGis

A new way of representing and communicating the 'manifestation' of the land has emerged since the introduction of digital devices, which are highly versatile and easily updated, particularly through the process of geolocation. In fact, parallel to Google's development of digital maps, geolocation technology and several digital mapping studies were developed [14] (p. 442). In recent decades, the approach to urban and territorial representation has been gradually changing. This approach offers the possibility of access to families of information responding to diversified needs, in order to express functions revealed by the interactive potential of digital technologies. Indeed, this new representation allows the user, by interacting with it, to partly become its creator [15] (pp. 14–16). This process, dynamic and interactive, is very different from the practices of the past, with a view towards breaking down the clear division between those who produce and those who read the map [16] (p. 1125).

The spatial data infrastructures aim to facilitate access to data through the web via network services, and to overcome problems related to the availability, quality, organization, and accessibility of data, providing new opportunities for planners, professionals in the field of environmental assessment, and public administrations at mainly local and regional levels [17] (pp. 12–17). Improving the quality and quantity of available information has the potential to positively influence the analysis of urban and environmental phenomena, offering opportunities for improved policy-making.

For a web GIS to be an effective tool capable of providing structured information and developing new forms of communication of the evolutionary phenomena of the urban landscape, it is essential that all the extremely heterogeneous data contained within the various documentary apparatuses be carefully organized. It is also important that the cartography be properly processed and georeferenced in such a way that each point belonging to different maps is uniquely determined within a predetermined reference system. As Bishop and Lange state in Visualization in Landscape and Environmental Planning [18], significant strides have recently been made with regard to computer graphics employed to visualize our environment in three or four dimensions, especially resorting to the use of animations useful for recording different time phases or, more generally, introducing motion into the representation of space. The city of Parma has been moving in this direction for several years, providing itself with an absolutely effective consultation tool that sees the possibility of superimposing the various planning instruments in force by means of interactive maps that, operating according to the logic of information systems, link to georeferenced spatial graphical data information of another nature (in this case, regulatory information), making it possible to obtain integrated information on what the planning instruments provide for (Figures 24–26).

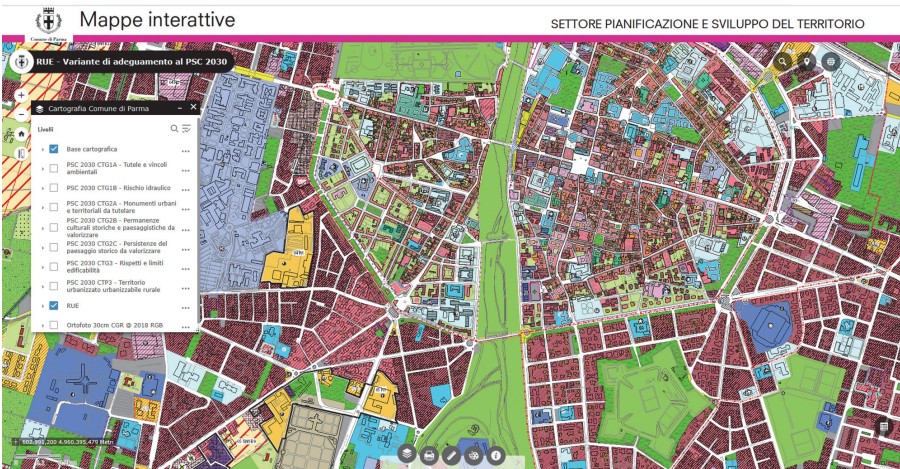

**Figure 24.** Data from 2023, Interactive maps: cartographic map and RUE.

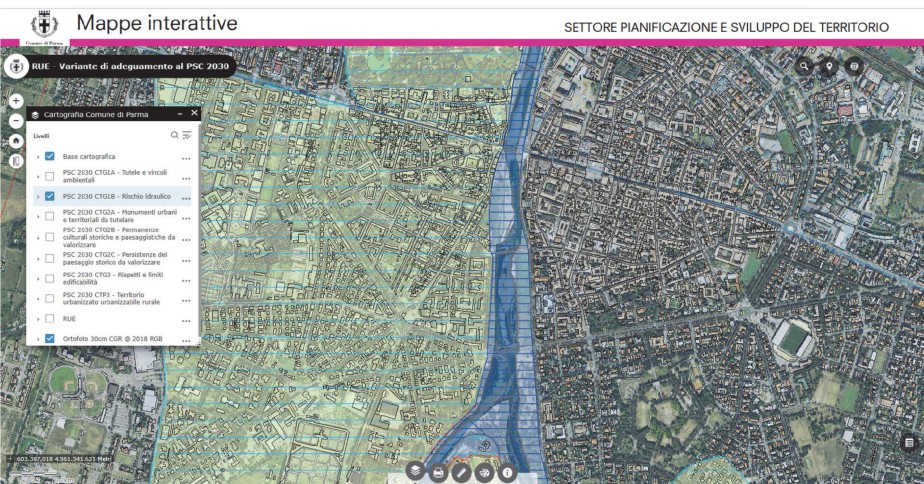

**Figure 25.** Data from 2023, Interactive maps: cartographic map, orthophoto, and PSC 2023 CTG1B hydraulic risk.

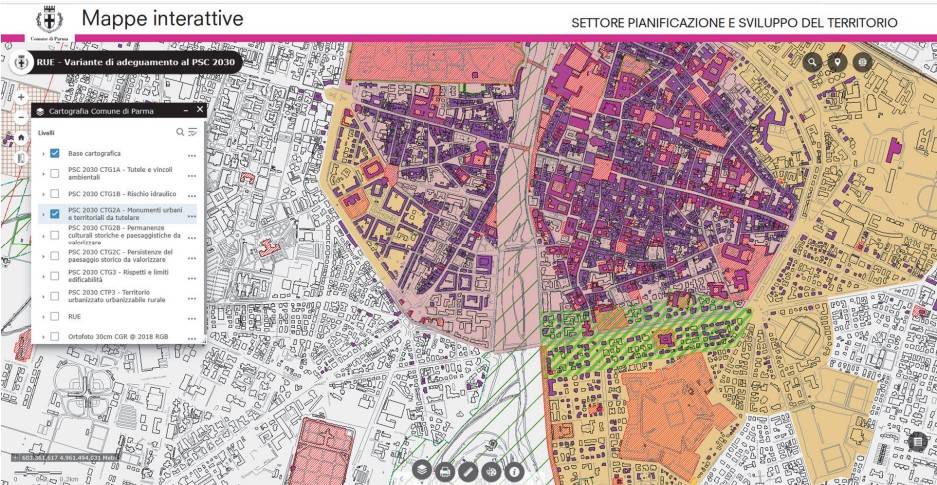

**Figure 26.** Data from 2023, Interactive maps: cartographic map and PSC 2023 CTG2A urban and territorial monuments to be protected.

This not only allows for accurate, real-time analysis but also an understanding of the potential to be increased and the vulnerabilities to be improved. The use of interactive maps on the website of Parma allows a dynamic analysis to capture the reality based on an interscalar subdivision of the territory; that is, from the synergy of the different levels of intervention. In fact, the reference system allows an overlay of information from the base cartography that can be easily queried (e.g., the different features, restrictions, or land registry information). This also speeds up the process of urban analysis by allowing high-resolution prints with associated legends, enabling functions linked to surveying or indeed accurate querying of the different features present. Therefore, these web GIS maps are tools that enable users to visualize, analyse, and understand geographic data in an interactive and intuitive way [19]. They represent a visual user interface that allows users to explore and interpret geographic information effectively.

A strong point is to try to substantiate and program the interactive maps of Parma like an actual atlas, oriented to the design representation of the territory, departing from the general meaning of "neutral" information system, which only apparently offers "objective" outputs, while it can be seen how the degree of interpretation and "subjectivity" is definitely higher in these environments than in traditional historical cartographies, whose strong interpretative charge is now taken for granted [20].

The use of the territory, analysed from the point of view of technologies, leads us to highlight some aspects. The possibility of establishing a different relationship with the workplace (the area of interest or the case study), based on the possibility of obtaining, consulting, and exchanging information at a distance through dynamic information that is no longer static, changes the performance of many services that use the network as a place of consultation and execution [21].

Interactive maps are well suited to interpret the need to be able to read diachronically and synchronously the complexity that today's city expresses, also through representations that, precisely through the sign that tends from symbol to ideogram not of meaning but of parametrization, express well the need for flexible perspectives of transformation able to adapt to the speed of change and the changing needs that today's city constantly expresses.

## 5. Conclusions

The complexity of the subject has been dealt with by means of a wide typology of expressive registers, declined according to specific needs, and in some cases even inventing visual languages created ad hoc to better meet the needs of recreating the whole multitude of elements that converge in a piece of landscape. This has been the case, for example, with thematic maps: they are articulated at the level of content, synthetic and descriptive, according to a logic of symbols and abstractions, but far from the immediacy of perception and, for this reason, quite complex to understand for an interlocutor who is not an expert. With this new way of understanding urban planning, there corresponds another type of using representation with ideogrammatic symbolism, the meaning of which differs from the plans designed in earlier years.

Urban and spatial analysis, usually preliminary, has developed its own codified graphic languages for understanding the phenomena that structure the city [22,23] (p. 49). An example is the process of unifying symbols and conventional signs to construct a common graphic language in urban planning, with the aim of obviating personal and improvised use. In fact, the floor plans must not leave any doubts of interpretation; the transmission of information is necessarily codified and unified: ‹‹Such graphic languages are still used today for the coding of zoning on a territorial and urban scale in which each pre-established colour corresponds to a precise homogeneity of function and to each sign a single reading of meaning. Floor plans leave no possibility of interpretation, the transmission of information is codified and unified [22,24]. Similarly, pre-prescriptive analysis charts use symbols and colours to bring together information through the use of graphical signs. For example, maps for ethnographic reading at different scales (national, territorial, and urban) become formidable tools for analysing complex contexts in which boundaries are difficult to trace›› [25] (p. 24).

A map, as a visual tool that makes use of various graphic languages, can perform a variety of functions. It is precisely the function of planning that is one of the most peculiar characteristics of maps. It should be recalled that every map is a project, first and foremost, a graphic project that is implemented reality, which changes through the act of representation [26]. It can be observed how, during this half century, signs have taken on a very different task: from the simple indication of the expansion to the identification of the functional zoning, which over the years has become more detailed and specified through the use of indices and standards, enriching the graphic apparatus of the increasingly articulated and complex set of rules. The last step is related to the use of signs in an ideogrammatic way, delivering them from the relationship with indices and standards, producing a flexibility that is now necessary for the implementation of urban planning [27] (p. 27). The crucial theme in the planning documents, precisely in relation to the scales at which they are expressed, is more than the relationship between signifier and signified, never fully resolved but above all continually in evolution. This is an issue which overlaps with the theme of the tools through which plan cartographies are expressed, never more crucial than in this case in bringing back the complexity of the vision, necessarily iterative and multi-scalar [28].

From static maps, used for centuries, there has been an evolution to dynamic and interactive maps that from GIS have led to web GIS—thanks to which the different themes of the different urban cartographies are put in relation to each other, with base maps, orthophotos, and digital technology. This evolution has made the passage to current systems possible and has allowed interactions between different information to become fluid: "These are mapping processes that are traced back to visual artifacts that give a graphic sign the one-to-one correspondence of a much more complex reality [29]. A process similar to that of GIS environment, in which the meaning of objects is defined by the attributes associated with a database that allows the interpretation and analysis of information on a geographical basis. These graphic expedients allow quickly, immediately and intuitively a variety of information that allows in the design phase, allows to evaluate the different options adopted for a place." [24] (p. 24). Cross-format relational databases and geographic information systems (GIS) play an important role in this process because they can be used to store, relate, and analyse alphanumeric data, maps, and images. They design and develop freely accessible tools that can also be used to reconstruct the memory of places [30,31].

**Author Contributions:** Conceptualization, C.V.; methodology, C.V.; formal analysis, C.V. and C.F.; investigation, C.V. and C.F.; resources, C.V. and C.F.; data curation, C.V. and C.F.; writing—original draft preparation, C.V. and C.F.; writing—review and editing, C.V. and C.F.; visualization, C.V. and C.F.; supervision, C.V. and C.F.; project administration, C.V.; funding acquisition, C.V. All authors have read and agreed to the published version of the manuscript.

**Funding:** This research received no external funding.

**Institutional Review Board Statement:** Not applicable.

**Informed Consent Statement:** Not applicable.

**Data Availability Statement:** Not applicable.

**Acknowledgments:** A heartfelt thanks goes to Arch. Alessandra Gravante for sharing the material on the http://archiviterritoriali.comune.parma.it/portal, accessed on 16 June 2023, on whose creation she worked actively under the guidance of Michele Zazzi of the University of Parma, in agreement with the Municipality of Parma. Equally heartfelt thanks go to the designers of the new PUG and the Plan Office of the Municipality of Parma for the competence and professionalism with which they worked on the formation of the new urban planning instrument. Thanks are also due to all the institutions, associations, and citizens who, with great interest and passion, contributed to the debate within the participation processes implemented.

**Conflicts of Interest:** The authors declare no conflict of interest.

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
