# Peer review of "Land Development Planning: New Principles and New Representations in the General Urban Plan of Parma"

_sustainability, doi:10.3390/su151914485_

Round 1

Reviewer 1 Report

The paper is like a review article. But there are some technical issues. Scientific soundness should be provided. Therefore authors should extend the discussion part. Use more articles to provide a scientific presentation.  

Author Response

Response to Reviewer 1 Comments

Point 1: The paper is like a review article. But there are some technical issues. Scientific soundness should be provided. Therefore authors should extend the discussion part. Use more articles to provide a scientific presentation.

Response 1: The paper is intentionally organized like a reviw article, because it refers to a case state, the General Urban Plan of Parma, to investigate the role of Representation in Urban Planning tools. Related to this new way of understanding urban planning is a different way of using representation using an ideogrammatic symbology whose meaning is different from the plans drafted in previous years.

This is the focus on which the contribution concentrates, which by retracing as in a chronological report the characteristics of the previous plans of the city of Parma, concentrates on this specific aspect, highlighting precisely the relationship between the meto-dological and conceptual approach of the new PUGs and their graphic representation.

However, for the revision of the contribution, greater scientific rigor was punctually detailed throughout the text and the references were also expanded. In fact, more specific and more recent references have been included, but also the sitography used.

The revised contribution with the parts in red has been uploaded as an attachment.

Reviewer 2 Report

The paper is focused on the analysis of the Emilia-Romagna Regional Law No. 24/2017 and its contents as an opportunity to analyse the tools through which the Governance of the Territory is implemented, from GIS to webGIS, and the changes in the graphic language through which the planning principles are expressed. The specific reference is to the Plan tools of Parma.

The topic is consistent with the topics of the Journal and the paper fits in the Sustainability journal Special Issue on “Visualising Landscape Dynamics” as it focuses on land development and contents representation through maps, landscape dynamics representation and visualization, and governance’s landscape design process, including the application of digital tools such as webGIS. The contribution is of interest at disciplinary level.

- It is suggested to better focus the abstract;

- It is suggested to change throughout the paper “Emilia Romagna” into “Emilia-Romagna” (from a quick search it seems to be the correct wording, possibly to be verified);

- please consider rewording lines 30-45, at least dividing into two paragraphs by putting a full stop at the line 40 (the semicolon is not used in English);

- the introduction should be more detailed. Although the central topic of the paper is rather specific, it would be useful to outline the research context and include the main bibliographical (and sitographical) references, also in relation to graphic language for cartographic representation, participatory planning topics, GIS and webGIS applications, related studies, etc.;

- Lines 59-66: the topic of urban planning archives consultation, and the need for digitization and dissemination devices, is very relevant, and of great interest for current disciplinary research lines. Also in this case, an overview of related works and state of the art would be need to support the discussion.

Just some suggestions:

M. E. Ruggiero and R. Torti, “Landscape Drawing and Comprehension: the Virtuous Passage of Scale in Digital Representation”, diségno, vol. 1, no. 5, pp. 59–68, Dec. 2019.

R. Marrocco, “Drawing, Measurement and Movement. The Representation of Space in Urban Maps (an Interdisciplinary Analysis)”, diségno, no. 7, pp. 151–164, Dec. 2020.

L. M. Papa, “Considerations about Old Maps in the Digital Era”, diségno, vol. 1, no. 5, pp. 91–102, Dec. 2019.

Bentley, F., Cramer, H., Hamilton, W., & Basapur, S. (2012, May). Drawing the city: differing perceptions of the urban environment. In Proceedings of the SIGCHI Conference on Human Factors in Computing Systems (pp. 1603-1606).

- figures are clear and effective; Figures 21, 23, 25 could be displayed smaller, or grouped in a single image;

- The bibliographical cited references are relevant to the research; anyway, references are limited in number. It is suggested that the bibliography be expanded to include national and international studies as well (on analytical methodologies / maps graphic representation / WebGIS applications / main topics of the research);

- as a minor issues, please check formatting according to the template (e.g. line 334).

- Citations of references in the text should be numbered in order of appearance.

Overall, the study is interesting and well structured; just minor revisions are suggested (as listed above) to improve the research framework, and the formatting.

A general check and revision of the English language is recommended, and long sentences should be divided.

Author Response

All suggestions were accepted and can be found in red in the attached paper. All linguistic and syntax suggestions have been corrected. In general, the part on disussion has been expanded with more references (even with some of those indicated). Some figures have been revised and regrouped. Specifically, provision was made to make the part of the introduction, the part related to digital maps and conclusions more substantial. The objective and some specific choices of the case study were better specified.

Reviewer 3 Report

I was positively surprised when I saw a theoretical paper concerning land use planning. After reading it I can say that it lacks theoretical part. I struggled to find the main aim of the paper. I feel as though I am chasing the essence of the paper till the very end but not finding it. The paper does have some very interesting information and good ideas in general, but it needs to be conceptualized better in order to have a clear goal, discussion and conclusion. I recommend focusing on the one segment which will be better theoretically presented and thoroughly examined in example of Parma. This way better conclusions can be made, and the research would have much more usefulness.

Author Response

Response 1: The paper is  organized like a reviw article, because it refers to a case state, the General Urban Plan of Parma, to investigate the role of Representation in Urban Planning tools. Related to this new way of understanding urban planning is a different way of using representation using an ideogrammatic symbology whose meaning is different from the plans drafted in previous years.

This is the focus on which the contribution concentrates, which by retracing as in a chronological report the characteristics of the previous plans of the city of Parma, concentrates on this specific aspect, highlighting precisely the relationship between the meto-dological and conceptual approach of the new PUGs and their graphic representation.  Specifically, provision was made to make the part of the introduction, the part related to digital maps and conclusions more substantial. The objective and some specific choices of the case study were better specified.

However, for the revision of the contribution, greater scientific rigor was punctually detailed throughout the text (and the references were also expanded.

The revised contribution with the parts in red has been uploaded as an attachment.

Reviewer 4 Report

In this paper, authors try to provide new insight and perspective into urban planning, with detailed time-track change analysis and variation trends/principle for the case region in Parma city. The manuscript is informative and the analysis is well-grounded with sufficient evidence in both pictural and theoretical regards. Some suggestions for further improvement are as follows.

1. Refine the title with more specific information for the studied case

2. Re-organize the whole content in a more scientific way. The present form is more like a case report, rather than an academic article.

3. What is the main research gap and the objective. Clarify the main study focus based on the background.

4. Clarify the novelty of this paper, since it is entitled by "new principle and representation". What are the key new elements? The used method and analysis approach are all off-shelf handy ones. What makes the chosen region a typical or representative case? 

5. Compare the results or conceptions with other similar studies in urban planning or city regeneration worldwide. How about the key advance or progress over available research.

6.  Extend the literature review substantially, with more recent and important publications. The present citations are somehow general, without lose relationship with current specific research. Moreover, most references were almost published more than 5 years ago. Please significantly update and state the recap and limitations of present studies.

7. Please re-organize the conclusion part. It is highly suggested to list main findings into several short bullets, with responses to the research objective and questions mentioned in the introduction.

8. Re-structure the PRO050 part, in a more inductive order and manner. Some pictures and figures seem redundant. Maybe the main stuffs need further compile with classifications and refinement.

The language and expressions need further polishment, especially by native speakers. Avoid typing errors.

Author Response

The paper is intentionally organized like a reviw article, because it refers to a case state, the General Urban Plan of Parma, to investigate the role of Representation in Urban Planning tools. Related to this new way of understanding urban planning is a different way of using representation using an ideogrammatic symbology whose meaning is different from the plans drafted in previous years.

This is the focus on which the contribution concentrates, which by retracing as in a chronological report the characteristics of the previous plans of the city of Parma, concentrates on this specific aspect, highlighting precisely the relationship between the meto-dological and conceptual approach of the new PUGs and their graphic representation.

However, for the revision of the contribution, greater scientific rigor was punctually detailed throughout the text and the references were also expanded.

The title was more specified and referred to the case study. Some images have been revised and grouped. In order to provide more rigor to the contribution, the inception part has been extended with reference to specific literature. The case study of Parma was chosen because it is one of the first municipalities, besides Modena and Bologna, to apply this law. In addition, it is a case study that was able to study closely and participate in participatory tables with citizens. For all these reasons, the conclusions were also applied and some concepts were more specified. The literature specifically, due to the specificity of the topic, focuses more on issues related to the digital planning system (GIS) and how this required new principles and representative processes.

The revised contribution with the parts in red has been uploaded as an attachment.

Round 2

Reviewer 1 Report

The manuscript upgraded with the revisons.

Please simplify the sentence a little more "The map, by its sign interaction, is capable of creating and conveying meanings that 112 depend on the historical and social context in which it was produced, and on the inter-113 preter, i.e., the interpreter's intended use of the cartographic elaboration, beyond the pur-114 pose for which it was made [8] (p. 91)."

  •  

For giving detail knowledge about GIS and land use cite article "https://doi.org/10.1007/s13762-020-02869-9"

Author Response

We reviewed the full manuscript. In particular, corrected some spelling errors and also the sentence indicated in lines 115-118. We also found the reference you pointed out helpful. We have also introduced the reference suggested with the number 19.

Thank you

Reviewer 3 Report

There has been some improvement in the paper but almost all my previous comments still stand. I think that this is not a suitable journal for this type of paper. As a review paper it should be better written to get better conclusions and much stronger discussion. There is a good basis for a great paper, but this form is not a good representation of this kind of research. Every story, even the scientific ones, needs a good, clear and strong conclusion to be a good one. If authors make clear goal of the research and follow a clear path to the conclusions this paper could be very useful for scientific community.

There are minor syntax errors.

Author Response

Dear,

thanks for the very helpful suggestions you have made in the review process, which we have tried to incorporate as far as possible. We would like to clarify one issue that is important to us. The paper deals specifically with the theme of Representation for Urban Planning and, only instrumentally to the theme of Representation, some principles of Planning are invoked. The call in this issue of the journal, focusing on the theme of Visualization for Planning dynamics had seemed to us to be relevant to the content of the paper, which deals precisely with this aspect (Visualization) and not with Planning.

Therefore, the paper deals with a dissertation of the methods of Visualization used, over time, in the specific case of Planning tools referred to the city of Parma. The reference to the planning approach has been invoked in an absolutely instrumental way to this topic, which even in our discipline is still relatively little studied still leaving the way open for new insights and research.

Kind regards

Reviewer 4 Report

Authors have made significant revisions and improvement according to the last comments. I suggest it to be accepted for publication after some minor corrections.

Authors are highly advised to double-check the whole content, since there are several typing errors and font/format inconsistency throughout.

Subsection heading missing? Only 3.1.1., where is 3.1.2?

Conclusions a bit lengthy.

Acceptable

Author Response

We have carefully corrected all the text. In doing this we also reformulated some sentences. The paragraphs have been revised. Thanks for the attention